

# Public awareness of seafood mislabeling

Savannah J. Ryburn[1], Wilker M. Ballantine[2], Florencia M. Loncan[2], Olivia G. Manning[2], Meggan A. Alston[2], Blaire Steinwand[2,3] and John F. Bruno[2]

[1] Environment, Ecology and Energy Program, The University of North Carolina at Chapel Hill, Chapel Hill, North Carolina, United States
[2] The Department of Biology, The University of North Carolina at Chapel Hill, Chapel Hill, North Carolina, United States
[3] Department of Zoology, The University of British Columbia, Vancouver, British Columbia, Canada

## ABSTRACT

A substantial portion of seafood is mislabeled, causing significant impacts to human health, the environment, the economy, and society. Despite the large scientific literature documenting seafood mislabeling the public's awareness of seafood mislabeling is unknown. We conducted an online survey to assess the public's awareness and perceptions of seafood mislabeling. Of the 1,216 respondents, 38% had never heard of seafood mislabeling and 49% were only 'vaguely familiar' with it. After being provided the definition of seafood mislabeling 95% had some degree of concern. Respondents were the most concerned about environmental impacts caused by seafood mislabeling and the least concerned about the social justice implications. Respondents who were also more concerned and familiar with seafood mislabeling stated that they would be more likely to purchase seafood from a vendor where the labeling was independently verified.

## INTRODUCTION

Seafood mislabeling occurs when a seafood product is labeled under an inaccurate species name, weight, geographic origin, or other characteristics. Mislabeling can take place during any stage of the seafood supply chain and can be intentional or unintentional making it difficult to pinpoint when and why products are mislabeled. Seafood mislabeling can be difficult for consumers to detect; filets can pass as a variety of species and even whole fish of different species can be nearly indistinguishable.

Seafood mislabeling is prevalent in both the international and domestic U.S. fish trade. In a recent study along the southeastern coast of the US, fish labeled as red snapper at sushi restaurants were mislabeled 100% of the time (*Spencer & Bruno, 2019*). Although, some species are rarely mislabeled and a recent meta-analysis found that the rate of mislabeling varied enormously among species, averaging 8% globally (*Luque & Donlan, 2019*). Information about seafood mislabeling is strongly skewed towards particular taxa and countries. Most study species are fish, leaving many invertebrates (*e.g.*, shrimp, scallops, crab) with insufficient data (*Luque & Donlan, 2019*). Fewer studies have been conducted in

Corresponding author
Savannah J. Ryburn, sryburn@live.unc.edu

countries such as China, Peru, and Thailand which are some of the top seafood producers, exporters, or importers (*Luque & Donlan, 2019*).

The global demand for seafood is at a record high with approximately 4.5 billion people depending on seafood for nutrition and livelihood (*Béné et al., 2015*). The global seafood trade is also valued at $143 billion (*FAO, 2018*) and 40% of the world's seafood is traded internationally (*Tveterås et al., 2012*). The seafood industry's development and complexity has outpaced the ability of government and the seafood industry to successfully monitor each step of the supply chain.

Seafood mislabeling compromises the consumer's ability to monitor dietary choices which could pose a multitude of public health concerns. Approximately 6.6 million Americans are allergic to some seafood (*Sicherer, Muñoz-Furlong & Sampson, 2004*). Some seafood can contain heavy metals and toxins and when eaten in large quantities can cause serious health issues, especially for pregnant women and children (*Marko, Nance & van den Hurk, 2014*). For example, escolar which contains high levels of zinc, was found mislabeled as "cod" in Hong Kong, resulting in over 600 people falling ill (*Jacquet & Pauly, 2007*). The flesh of higher trophic level fishes (*e.g.*, tuna and barracuda) can contain elevated levels of mercury or ciguatera toxins (*Matta et al., 1999*), and consumers are put at risk when these species are substituted for lower trophic level fishes.

Mislabeling is also commonly used to mask the sale of illegally caught species which can lead to the overexploitation of at-risk species (*Spencer et al., 2020*). Twenty six million tonnes of global seafood catches are estimated to be from illegal or unreported fishing (*Agnew et al., 2009*). In a study conducted in Brazil, 24 out of 44 (55%) shark samples purchased from the seafood markets were actually found to be endangered largetooth sawfish (*Melo Palmeira et al., 2013*).

Additionally, there is a strong economic incentive to intentionally mislabel seafood, selling species with lower wholesale costs as a more profitable one (*e.g.*, selling tilapia as red snapper) due to the substantial price difference. Red snapper and grouper are high value and among the most commonly mislabeled fish, with price differences of up to 244% between the genuine and fraudulent fish (*Stiles et al., 2013*). Since fish is one of the most-traded food commodities (*FAO, 2016*) and make up almost a quarter of the global intake of animal protein (*Naaum et al., 2016*), product substitutions can result in substantial economic impacts. According to one study the US economy could be losing seven million dollars a year as a result of species substitution (*Cline, 2012*).

Marine fisheries employ approximately 260 million people (*Teh & Sumaila, 2013*) making the seafood industry one of the world's largest employers (*Nakamura et al., 2018*). Up to 70% of the seafood export market is in developing countries. In seafood hubs such as Indonesia, Thailand, Vietnam, the Philippines, and Peru, slavery and child labor within the seafood industry are widespread (*Nakamura et al., 2018*). With the global demand for seafood consistently increasing, illegal, unreported, and unregulated fishing has ensued in slavery and human trafficked labor (*IPOA-IUU, 2001*).

Seafood mislabeling is commonly reported on in the media. Many large platforms such as National Geographic, CBS, CNN, and newspapers (*i.e.*, Los Angeles Times, Tampa Bay Times, Miami Herald, New York Times) have all reported on seafood mislabeling for

years. Yet the general public's awareness of mislabeling is unknown. There is a positive association between the public's knowledge of environmental impacts and environmentally conscious behavior (*Zabkar & Hosta, 2013*). A well-known example was the 2018 video of a scientist removing a plastic straw from a sea turtle's nostril. The video went viral and led to greater public awareness and a movement to phase out the use of plastic straws (*Figgener, 2018*). Cities such as Seattle, Washington and San Francisco, California placed bans or limits on plastic straws while companies such as Disney and Starbucks have phased out their usage. Public awareness motivates policy makers and changes in industry practices making it an important factor when addressing issues such as seafood mislabeling.

The purpose of this study was to determine how familiar and concerned people are about seafood mislabeling. We also determined how likely people would be to preferentially purchase seafood from a restaurant or retailer where the labeling was independently verified. Ultimately, this information was used to quantify the public's awareness of seafood mislabeling and evaluate interest for independently verified seafood certifications.

## MATERIALS AND METHODS

We conducted an online survey to measure the public's awareness of seafood mislabeling and whether they would be more likely to purchase seafood from a vendor where the labeling was independently verified. The survey instrument was developed *via* Qualtrics survey software, and included 10 questions (Supplement 1). Each question was presented on its own page and once the respondent had submitted the answer to a question they could not revisit or change their answer.

The first three questions were used to assess how often and where the respondent purchases seafood and their knowledge of seafood mislabeling. Once the respondent had answered question three, we then provided the definition of seafood mislabeling and two examples:

*Seafood mislabeling or fraud occurs when a seafood product is labeled under a different name than its actual species identity. For example, when farmed, imported shrimp is sold as "local, fresh" or when tilapia is sold as red snapper.*

Questions four and five assessed the respondent's concerns regarding seafood mislabeling (after learning the definition and becoming aware of the problem). The sixth question provided the statement: "About one third of all seafood is mislabeled. And for some species and vendor types, the percentage is much higher". The respondent was then asked how likely they would be to preferentially purchase seafood from a restaurant or retailer where the labeling was independently verified. The remaining four questions were designed to collect demographic information. This information included the respondent's year of birth, if they were a student, what country they resided in, and if they resided in the US what state. The study was conducted following human-research ethical standards. Respondents gave informed consent by voluntarily choosing to fill out the survey and their identity was not recorded. The study was evaluated by the Office of Human Research

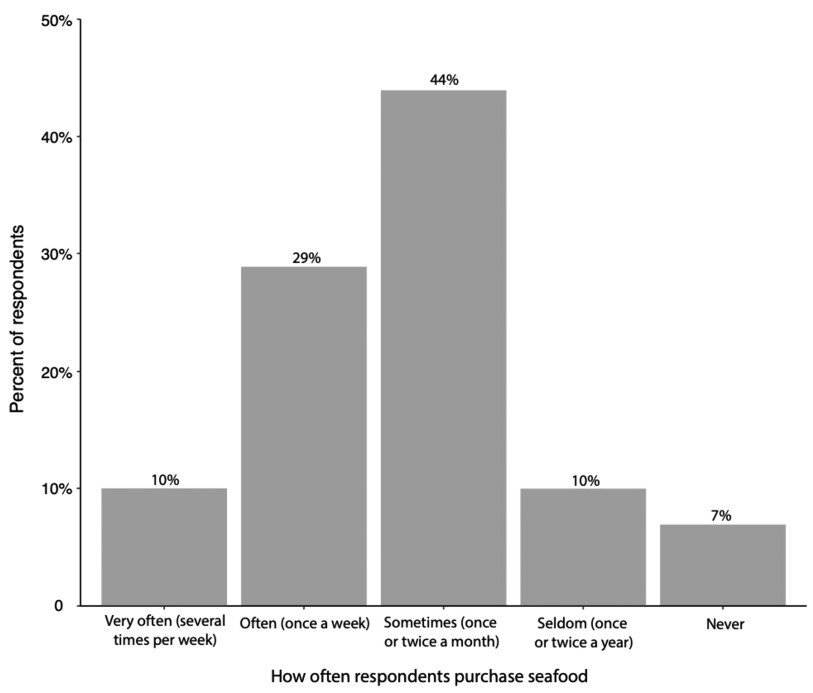

**Figure 1 How often respondents consume seafood.** Figure includes all responses (*n* = 1,216).

Ethics at the University of North Carolina at Chapel Hill and determined to be exempt from IRB authorization (IRB-19-2761).

The survey was active from October 2019 to March 2020, although the majority of responses occurred in October. The link to the survey was distributed as a shareable post on an array of social media platforms including Twitter, Facebook, and Instagram. Flyers which included the link and a QR code were also posted in local markets (*e.g.*, meat markets, fish markets, farmers markets, *etc.*) and healthcare facilities throughout major cities in North Carolina (*e.g.*, Chapel Hill, Charlotte, High Point, Raleigh, and Greensboro) and also in Miami and Seattle. These locations were chosen opportunistically. Most questions within the survey used Likert-type scale responses, and Chi square tests were used in R Studio to analyze the data. A total of 95% confidence intervals were calculated on reported percentages with JMP Pro 16.

## RESULTS

A total of 1,216 individuals completed the survey. Of these respondents 98% (95% CI [0.96–0.98]) were from the United States and 62% (95% CI [0.61–0.67]) resided in North Carolina (Supplements 3 and 4). A total of 40% (95% CI [0.37–0.42]) of the respondents were students. When asked "how often do you consume seafood?", 44% (95% CI [0.41–0.46]) of respondents answered 'sometimes' (once or twice a month) while 29% (95% CI [0.27–0.32]) responded 'often' (once a week) (Fig. 1). When asked "where do you typically purchase your seafood?", the majority of respondents answered at a 'grocery store' (40%) (95% CI [0.99–1]) or at a 'restaurant' (37%) (95% CI [0.99–1]),
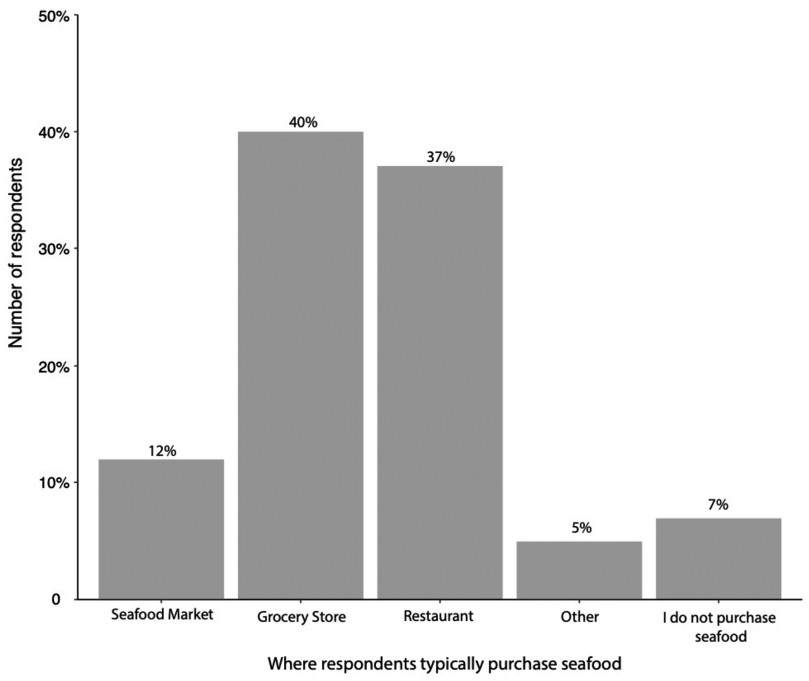

**Figure 2 Where the respondents typically purchase seafood.** Respondents were told to check all that apply. Figure includes all responses (*n* = 1,216).

**Table 1 Distribution of survey respondents according to how concerned and familiar they were with seafood mislabeling or seafood fraud.**

|  | Never heard of it | Vaguely familiar | Very familiar | Total |
|---|---|---|---|---|
| Not at all concerned | 43 (4%) | 15 (1%) | 3 (0.2%) | 61 (5%) |
| Slightly concerned | 96 (8%) | 62 (5%) | 9 (0.8%) | 167 (14%) |
| Moderately concerned | 166 (14%) | 233 (19%) | 38 (3%) | 437 (36%) |
| Very concerned | 92 (7%) | 188 (16%) | 57 (5%) | 337 (28%) |
| Extremely concerned | 65 (5%) | 95 (8%) | 54 (4%) | 214 (17%) |
| Total | 462 (38%) | 593 (49%) | 161 (13%) | 1,216 |

**Note:**
The respondents' level of concern was assessed after they were provided the definition and an example of seafood mislabeling.

while 12% (95% CI [0.98–1]), 5% (95% CI [0.96–1]), and 7% (95% CI [0.97–1]) responded 'seafood market', 'other', or 'I do not purchase seafood' (Fig. 2).

## The public's familiarity and concern for seafood mislabeling

Of the 1,216 respondents 13% (95% CI [0.11–0.15]) said they were 'very familiar' with seafood mislabeling or seafood fraud, 49% (95% CI [0.46–0.52]) were 'vaguely familiar', and 38% (95% CI [0.35–0.41]) had 'never heard of it before' (Table 1). After being provided the definition of seafood mislabeling 17% (95% CI [0.16–0.20]) of respondents said that they were 'extremely concerned' about seafood mislabeling, 28% (95% CI [0.25–0.30]) 'very concerned', 36% (95% CI [0.33–0.39]) 'moderately concerned', 14%

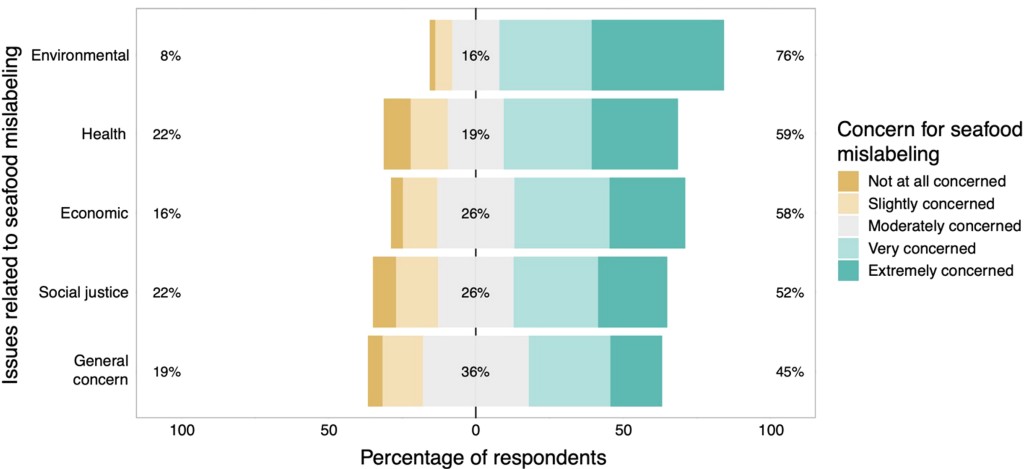

**Figure 3 The public's response to the questions: "How concerned are you about seafood mislabeling?" (*i.e.*, general concern) and "Which issues related to mislabeling concern you?".** The percentages on the left side of the figure include responses for both "Not at all concerned" and "Slightly concerned" while the percentages on the right include responses for "Very concerned" and "Extremely concerned". The percentages in the middle include responses for "Moderately concerned". Figure includes all responses ($n = 1,216$).

(95% CI [0.12–0.16]) 'slightly concerned', and only 5% (95% CI [0.04–0.06]) were 'not at all concerned'.

When asked about their concern for specific issues related to seafood mislabeling, respondents had the most concern for environmental issues associated with seafood mislabeling and the least concern for social issues associated with seafood mislabeling. More than 3/4 of respondents were either 'very concerned' or 'extremely concerned' in regards to environmental issues whereas only 52% of the respondents were very or extremely concerned about social justice issues (Fig. 3).

Respondents were categorized by familiarity and general concern for seafood mislabeling. The majority of respondents had never heard of seafood mislabeling ($n = 462$) or were vaguely familiar with seafood mislabeling ($n = 593$). The majority of respondents were also moderately concerned with seafood mislabeling ($n = 437$). Respondents who were more familiar with seafood mislabeling were significantly more concerned about it (Table 1, Fig. 4, *Chi-square* = 109.59, *df* = 8, *p* < 0.0001).

## If the public is more familiar and concerned with seafood mislabeling will they be more likely to purchase seafood from a verified source?

The majority of participants responded that they would either be 'somewhat likely' (36%) (95% CI [0.34–0.39]) or 'extremely likely' (38%) (95% CI [0.35–0.41]) to preferentially purchase seafood from a restaurant or retailer where the labeling is independently verified. When compared to the respondent's concern for seafood mislabeling, the likelihood to preferentially purchase seafood from a restaurant or retailer where the labeling is independently verified significantly increased with the level of concern (Fig. 5A, *Chi-square* = 100.73, *df* = 4, *p* < 0.0001). When compared to the respondent's familiarity with seafood mislabeling, the likelihood to preferentially purchase seafood from a

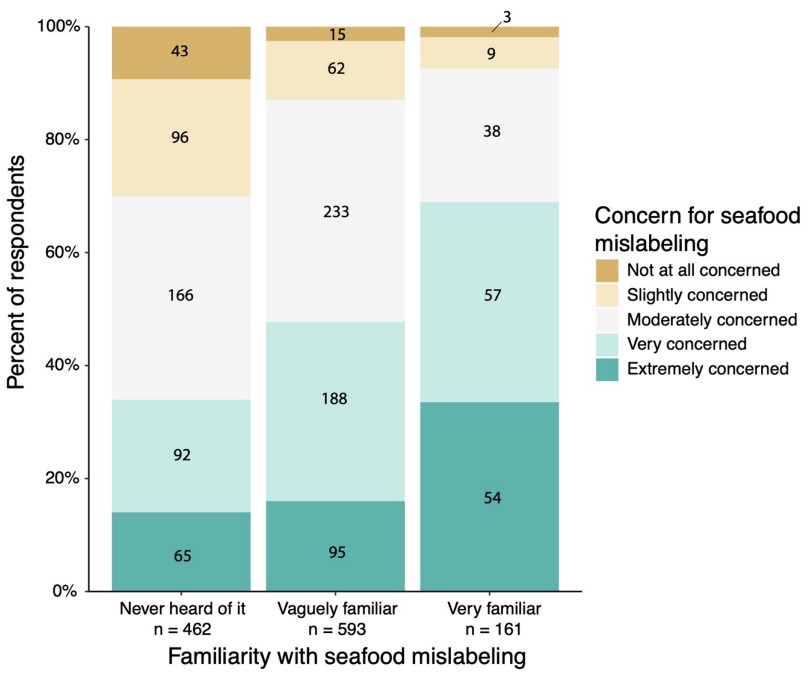

**Figure 4 Distribution of survey respondents based on their familiarity with seafood mislabeling in relation to how concerned they were with seafood mislabeling.** Numbers overlayed on the figure represent the number of respondents within each category. Figure includes all responses ($n = 1,216$).

restaurant or retailer where the labeling is independently verified also significantly increased with the level of familiarity (Fig. 5B, *Chi-square* = 75.85, *df* = 8, *p* < 0.0001).

## DISCUSSION

Although seafood mislabeling has been documented since 1915 (*Golden & Warner, 2014*) and is commonly reported on in the media, 38% of respondents had never heard of it before completing the survey. However, it is important to note that our survey respondents were skewed towards younger people and residents of North Carolina. Thus, more intensive sampling in other regions of the U.S. or of different demographics could yield different results. Our results indicated that respondents who were more familiar with seafood mislabeling were significantly more concerned about its impacts. Respondents had the greatest concern for environmental issues associated with seafood mislabeling. Over recent decades consumers have become increasingly more environmentally conscious (*Zabkar & Hosta, 2013*), especially when purchasing food therefore buying more organic and locally sourced products. Environmental issues such as overfishing, pollution, and illegal fishing have dominated what the general public deems as sustainable seafood and has determined what seafood to avoid. Most people are far removed from the fishing industry and unaware of the processes through which seafood makes its way from to grocery stores and restaurants. Social justice issues such as seafood slavery do not have a direct impact on as many people as economic or health issues caused by seafood mislabeling. During the survey it could have been difficult for respondents to visualize
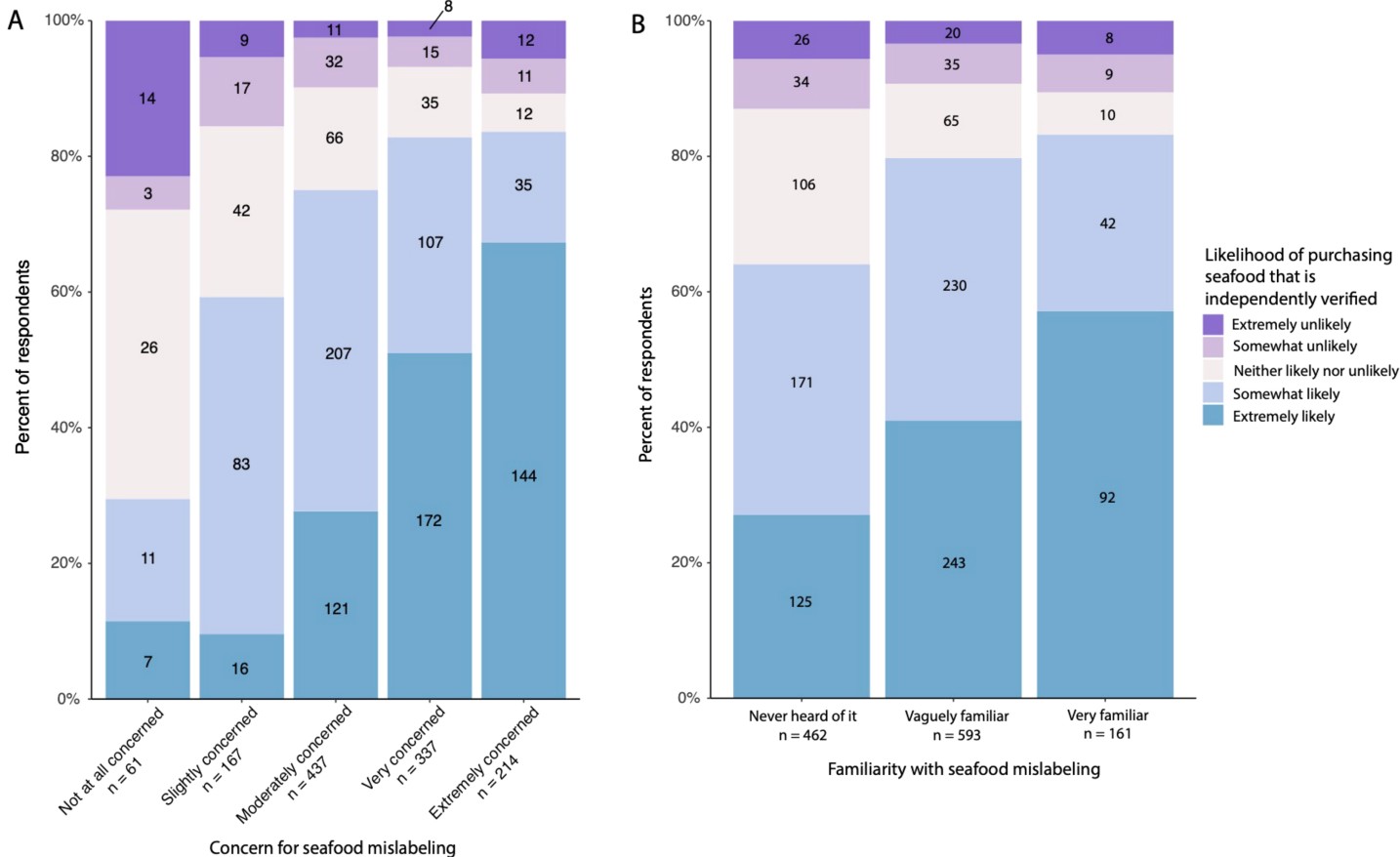

**Figure 5** Distribution of survey respondents based on their likelihood of preferentially purchasing seafood from a restaurant or retailer where the labeling is independently verified in relation to (A) how concerned they were with seafood mislabeling or (B) how familiar they were with seafood mislabeling. Numbers overlayed on the figures represent the number of respondents within each category. Each figure includes all responses (*n* = 1,216).

what social justice issues and seafood slavery actually are without previous knowledge and/or further explanation. This could be why respondents showed the least concern for social justice issues associated with seafood mislabeling.

When scientific investigation of seafood mislabeling was accompanied with public outreach in Europe, the mislabeling frequency of Atlantic cod decreased (*Mariani et al., 2015*). Outreach tools other than news sources (*i.e.*, newspapers and television) that consumers rely on to make sustainable choices include verified labels (*e.g.*, OceanWise and the Marine Stewardship Council (MSC)). Such labels are based on conservation status and fishing practices, but can be vague (*Jacquet & Pauly, 2007*). Many state "certified sustainable seafood" with no further explanation for consumers. MSC also commissions DNA tests on its certified products to verify the authenticity of their labeled seafood (*Barendse et al., 2019*). Due to these tests seafood certified by MSC is 3–5 times less likely to be subject to harmful fishing practices (*e.g.*, mislabeling) than uncertified seafood (*Gutiérrez et al., 2012*).

Although multiple independently verified seafood labels exist, many focus their DNA verification on a limited number of species and a large number of consumers may not even know that these labels exist. Our results indicate that consumers who were more familiar and/or concerned about seafood mislabeling would preferentially purchase seafood from an independently verified source. Therefore, increasing the general public's education on seafood mislabeling as well as increasing the amount, scope, and public knowledge of independently verified seafood labels is necessary. A potential approach to increasing awareness is through social media campaigns on platforms such as TikTok, Instagram, YouTube, *etc.* Platforms such as these reach a wide demographic of people all over the world and can educate individuals with either an infographic or short video clip. With an increase in consumer knowledge, the demand for verified seafood will become greater therefore seafood suppliers will be held more accountable for the quality of their products. The independent labels must provide links to detailed information about the verification process (*i.e.*, DNA testing) and their classification of a "sustainable fishery" for consumers. Seafood mislabeling is driven by consumer demand. Therefore, consumers can reduce seafood mislabeling by purchasing more verified products consequently mitigating the issues associated with seafood mislabeling.

## ACKNOWLEDGEMENTS

This project was conducted as part of the Seafood Forensics CURE class at the University of North Carolina at Chapel Hill.

### Funding

The project was funded by the Department of Biology at UNC-CH and funded by the QEP (Quality Enhancement Plan) CURE (Course-based Undergraduate Research Experience) initiative. The project was also funded by the National Science Foundation (OCE #1737071 to John F. Bruno). There was no additional external funding received for this study. The funders had no role in study design, data collection and analysis, decision to publish, or preparation of the manuscript.

### Grant Disclosures

The following grant information was disclosed by the authors:
Department of Biology at UNC-CH.
QEP (Quality Enhancement Plan) CURE (Course-based Undergraduate Research Experience) initiative.
National Science Foundation: OCE #1737071.

### Competing Interests

John F. Bruno is an Academic Editor for PeerJ.

## Author Contributions

- Savannah J. Ryburn analyzed the data, prepared figures and/or tables, authored or reviewed drafts of the article, and approved the final draft.
- Wilker M. Ballantine conceived and designed the experiments, authored or reviewed drafts of the article, and approved the final draft.
- Florencia M. Loncan conceived and designed the experiments, authored or reviewed drafts of the article, and approved the final draft.
- Olivia G. Manning conceived and designed the experiments, authored or reviewed drafts of the article, and approved the final draft.
- Meggan A. Alston conceived and designed the experiments, authored or reviewed drafts of the article, and approved the final draft.
- Blaire Steinwand conceived and designed the experiments, authored or reviewed drafts of the article, and approved the final draft.
- John F. Bruno conceived and designed the experiments, authored or reviewed drafts of the article, and approved the final draft.

## Human Ethics

The following information was supplied relating to ethical approvals (*i.e.*, approving body and any reference numbers):

The study was evaluated by the Office of Human Research Ethics and determined to be exempt from IRB authorization (IRB-19-2761).

## Data Availability

The raw data is available in the Supplemental File and at Figshare: Ryburn, Savannah; Ballantine, Wilker M.; Loncan, Florencia Maria; Manning, Olivia Grace; Alston, Meggan; Steinwand, Blaire; et al. (2022): Survey results for Ryburn et al 2022.xlsx. figshare. Dataset. https://doi.org/10.6084/m9.figshare.16679848.v1.

## Supplemental Information

Supplemental information for this article can be found online at http://dx.doi.org/10.7717/peerj.13486#supplemental-information.

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
