# Peer review of "Public awareness of seafood mislabeling"

_PeerJ, doi:10.7717/peerj.13486_

## Round 0.1 · original submission · Major Revisions

Both reviewers found this work potentially useful, and have a number of suggestions that I think will improve the manuscript. In particular, please pay attention to the comments of Reviewer 2 (adding additional statistical analyses, measures of uncertainty, etc).

Reviewer 1 ·

Basic reporting

The article is largely well written in English and uses clear and technically correct terms. Occasionally, the authors speak in sweeping all-inclusive statements and this phrasing should be reconsidered (example: Line 27, 'seafood awareness is unknown'). Similarly, from lines 101 to 104, the authors state seafood mislabeling is commonly reported through major media outlets, but then again state it is unknown by the public. The sweeping statements seem contradicted.

Introduction and background are provided, however, there are concerns with how well the literature was reviewed by the authors due to potential inaccuracies and citation of some references that don't seem to match the intent of the current paper. For example, Line 74, Willette et al. 2017 paper was not on toxins, heavy metals or human health risk, Ray 1989 speaks about slavery but not in the context of IUU fishing which I don't believe was coined as a term until more recently in the early 2000s (see FAO's IPOA-IUU), and some references can not be checked because they are incompletely cited (e.g. Stiles et al. 2013). Furthermore, other use of references are misleading. For example Line 61-63, the authors state few studies have been conducted in China, Peru, and Thailand, and cite a single review by Luque & Donlan 2019. This 2019 paper was extensive, however, the authors should also screen the literature themselves before making such a claim. Peru, for example, has a growing number of studies on seafood labeling, including from their government institutions such as IMARPE.

The article structure, figures and tables are professionally done. Raw data is shared & appreciated.

Experimental design

The research question is well defined. It is stated how it could fill the research gap of gauging public awareness of seafood mislabeling.

Although the concept of the study is solid, the implementation of the study seems narrow and at risk of bias due to the way the survey was conducted. A major concern is how survey participants were included. Details are somewhat vague and potentially skewed. For example, on lines 143-149, the authors state that it was shared via social media, but it is unclear how? Were paid advertisements put out or was the survey shared only through the authors individual social media accounts? Given the authors are at a university and state 40% of participants were students, this seems skewed. Additionally, 62% of participants were from 4 North Carolina cities that are all in-land, away from coastal cities where seafood issues may be more familiar. The authors state signs were posted, but in how many locations, and why only in local markets and healthcare facilities? Why choose these locations over fish markets, seafood restaurants, etc.? More details are needed on postings in Miami and Seattle, notably because only 4 respondents were from Washington State based on supplemental data. Further, based on the supplemental data provided, only 22% of participants were from coastal states (excluding respondents from NC). Does this matter that the survey seems to exclude coastal towns and cities?

Line 138-139 - Please more clearly explain what human-research ethic standards means, particularly because in line 141 the authors state it is exempt from IRB authorization.

Line 140-142 - Which Office of Human Research Ethics? UNC?

More generally, I feel the study is limited in its reach and I strongly encourage them to consider broadening the number of participants included. The authors cite a nice example of how to set up such a survey study as this by using Sicherer et al. 2004. Examples like this should be modelled from for increased rigor.

Validity of the findings

Major concern - Data lacks statistical analysis, but are largely reported as percentages only from the survey responses. This weakens the data. Rather authors should include 95% confidence intervals for all reported percentages. I find the lack of 95% CI particularly curious as it was a major issue in seafood mislabeling studies discussed by Luque & Donlan 2019 that the authors cite early in this manuscript.

Authors should more clearly and directly state the scope of the study and the limits of the experimental design. I would suggest the title be modified to speak to the geographic scope of the study area. Arguably states with very low respondent numbers (<10) could be dropped. A suggestion would be 'Public Awareness of Seafood Mislabeling in North Carolina' or '... in the Southwest United States'. As the authors state, major media outlets may indeed pick up on this study and sensationalize the results & conclusions without considering the limits of the research (majority of respondents from inland North Carolina, a plurality are students, surveys conducted through social media channels, etc.).

Additional comments

I do feel this could be an important contribution to the PeerJ readership and the larger scientific audience, however, the study lacks enough rigor and statistical foundation to have confidence in the conclusions. I feel with major revisions and re-focusing on the geographic scope of the study, it would be well suited for publication.

Reviewer 2 ·

Basic reporting

In my opinion, this work is clear and easy to read. I think the context shown in the introduction is correct and the objective is clear. The material offered (raw data, figures, and tables) is adequate and clear. I detected only a few errors (probably errata) that I have detailed in my comments.

Experimental design

I think the experimental design is good and the analysis and statistics were fine. However, there are some small issues that I think the authors should review before publishing. I add a few proposals that I miss in the analysis and/or results.
- L128-130: I do not agree with the definition of mislabeling. Mislabeling and fraud are not sinonims. Fraud implies intentionality. When a product is unintentionally mislabeled, it should not be called fraud. In the case of mislabeling, on many occasions, it is impossible (although it can be supposed) to ensure that there has been a fraud, except when it is impossible that the species had been captured at the same time and had not coincided during their supply chain.
On the other hand, I do not understand exactly the position of the definition in the survey. It is between Q3 and Q4. Does it mean you consider that Q3 was answered without knowing the definition? Did it appear after answering? Or the definition was to clarify the mislabeling before Q3 responses? This is important because knowing what is mislabeling (definition) sure will change the response of Q3.
- L132-136: The statement showed before the Q6 clearly conditions the response of the sixth question. If you do not use any statement there probably the participant's responses would probably be less inclined to attach such importance to independent verification of products. But if you tell them previously that mislabeling is enormous, they will obviously reply that they would prefer to be sure of what they are buying. This is good to know that if the population were aware of the problem, they would really value such verification and control. But in no case can it be taken as a reflection of current society when in the first questions of the test it is clear that the population is not aware that mislabeling is a problem. For the rest of the people (non-participants who have not this information), it is still an unknown risk and therefore they would not give so much importance to the independent validation. I think you need to explain better the intention of the question and the potential conclusions of the responses.
- L137: For future studies, it is also interesting to ask “what is the educational level” in order to know if it is correlated to the concern, knowledge, or interest about mislabeling or the news related with. I also miss some calculations about the relationship between age and the responses.

Validity of the findings

I think this study is useful and a great view of society's awareness about seafood mislabeling. The article shows that much work is still needed to do for spreading properly the knowledge about mislabeling and its important consequences. I think the authors should add to the discussion some aspects such as “who should be responsible for the control and validation tools”, etc. I have detailed these proposals in my general comments. But, despite this, this article has a lot of potentials, it is clear and direct and as a point to highlight, towards the end, the authors transform the lack of knowledge and awareness about mislabeling into a call to society to empower itself and be aware of mislabeling, saying that everyone, as consumers, can change the market and its failures. This hopeful message seems really important.

Additional comments

Introduction
- L51: “can be difficult to detect for consumers” instead “can be difficult for consumers to detect”
- L52: fillets
- L52-L53: several species are difficult to recognize by consumers: in some cases because the species are really similar and even for an expert is difficult but other times because many people do not know the natural appearance of the species, only in the market and many times only after any processing.
- L59: studied species
- L79: I miss some connector: “On the other hand”, …
- L85: The same, I miss some connectors as “In addition” or similar
- L89: Since fish IS …
- L101: Yes, seafood mislabeling is commonly reported on media (correct, you have two prepositions). But in my opinion, not enough compared with the real frequency of mislabeling currently. Maybe in the US seafood mislabeling is on regular media (not specialized) every day but in my country, we can see news about it once every 4 or 5 months hopefully. Even though it happens every day. It is because for me it is not surprising that people really not be aware of the huge problem that it is and the huge consequences that it has.

Results
- Table 1: The percentages are not the same as in the main text (L161-165). Please review and correct to have the same percentages in both places.
- Figure 3: Is the percentage in the medium of the bars the “moderately concerned”? Please add it to the figure legend.

Discussion
- L194: the percentage is not the same in the text (38%) as in the Table (37%). Please check it.
- L208-209: Another reason can be the people see the social justice issues as slavery as a problem really far from their country so … many times “if you do not see that it does not exist ...”.
- L210-211: I understand the sentence but I think the expression is weird. Please, review and rewrite it.
- L215: “states” instead “state”
- L212-2019: in addition, several times consumers do not know the verified labels available and their characteristics or requirements. That is another problem about people's awareness. There are sustainable certifications that none knows …
- L223225: So the development of more tools to identify more species and detect fraud is extremely needed.
- I miss some points in the final section of the discussion that maybe you can add (a few sentences). (1) Who should create the verified labels or verification systems? Governments, associations, volunteers, all, … All these systems mean money. (2) I miss some proposals to increase the general public’s education on seafood mislabeling (opinion or original proposal and comparison with some bibliography).

Supplements
- Supplement 2 graph: in the bar of 18-24 years old say 8-24. Please correct it.

---

## Round 0.2 · accepted · Accept

Thanks for the revisions, the point-by-point responses are clear and improve the manuscript.